# The dynamics of overlayer formation on catalyst nanoparticles and strong metal-support interaction

Arik Beck [1,2], Xing Huang [3✉], Luca Artiglia[2,4], Maxim Zabilskiy [2], Xing Wang[1,2], Przemyslaw Rzepka[1,2], Dennis Palagin [2], Marc-Georg Willinger [3✉] & Jeroen A. van Bokhoven [1,2✉]

Heterogeneous catalysts play a pivotal role in the chemical industry. The strong metal-support interaction (SMSI), which affects the catalytic activity, is a phenomenon researched for decades. However, detailed mechanistic understanding on real catalytic systems is lacking. Here, this surface phenomenon was studied on an actual platinum-titania catalyst by state-of-the-art in situ electron microscopy, in situ X-ray photoemission spectroscopy and in situ X-ray diffraction, aided by density functional theory calculations, providing a novel real time view on how the phenomenon occurs. The migration of reduced titanium oxide, limited in thickness, and the formation of an alloy are competing mechanisms during high temperature reduction. Subsequent exposure to oxygen segregates the titanium from the alloy, and a thicker titania overlayer forms. This role of oxygen in the formation process and stabilization of the overlayer was not recognized before. It provides new application potential in catalysis and materials science.

[1] Institute for Chemical and Bioengineering, ETH Zurich, Vladimir-Prelog-Weg 1, 8093 Zurich, Switzerland. [2] Laboratory for Catalysis and Sustainable Chemistry, Paul Scherrer Institute, Forschungsstrasse 111, 5232 Villigen, Switzerland. [3] Scientific Center for Optical and Electron Microscopy (ScopeM), ETH Zurich, John-von-Neumann-Weg 9, 8093 Zurich, Switzerland. [4] Laboratory of Environmental Chemistry, Paul Scherrer Institute, Forschungsstrasse 111, 5232 Villigen, Switzerland. ✉email: xing.huang@scopem.ethz.ch; marc.willinger@scopem.ethz.ch; jeroen.vanbokhoven@chem.ethz.ch

One of the most important class of catalysts, due to its high activity, is noble metal nanoparticles stabilized on oxide supports. They find use in a broad range of applications: in fuel cells, the treatment of exhaust gas, energy conversion, petrochemistry, and the production of fine chemicals[1,2]. In the early days of catalysis research, oxide carriers were considered to be mere inert carriers for highly dispersed metal nanoparticles. Forty years ago experiments with noble metal-titania catalysts[3–5] demonstrated that oxide supports are far from inert in such catalytic systems and, therefore, the phenomenon was named the strong metal-support interaction (SMSI). SMSI was observed in noble-metal catalysts supported on reducible oxides such as titania. It manifests itself as a profound loss of capacity to chemisorb molecules and intermediates, such as carbon monoxide and hydrogen, after high-temperature reduction. Because the adsorption behaviour of molecules on catalysts is a key factor in determining the activity and selectivity, this phenomenon is of paramount importance[6–8]. Typically, noble metal catalysts exhibit very high activity but suffer from poor chemoselectivity. Tuning such catalysts by a high-temperature reduction treatment, revealed impressive improvement of the chemoselectivity without loss in catalytic activity[9–11]. These reported advances nevertheless lack of an actual understanding how the reductive pretreatment alters the catalyst, but often mainly refer to the original work[5,12]. The studies suggested two possible origins of the SMSI state: (i) During high-temperature reduction, a bimetallic alloy[12,13] forms between the metal and the metal component of the support. This may result in a reduction of chemical adsorption strength. (ii) Alternatively, the support, in a partially reduced state, migrates onto the metal during reduction, thus sterically hindering adsorption on the metal. The spectroscopic study of this phenomenon is challenging, especially with regard to the fact that the bulk properties of both the support and the supported metal do not change upon reduction, because the SMSI process mainly involves the surface of the material. Therefore, in the 1990s, surface science based on model systems[14] and electron microscopy of the actual catalysts[15] made it possible to observe coverage of the metal by the support species as the origin of the suppressed chemisorption. Since then, there has been general agreement that partially reduced oxide species migrate onto the nanoparticle surface of the metal and, thereby, lead to (partial) encapsulation[16]. The process is believed to be reversible by means of high-temperature oxygen treatment to re-oxidize the reduced species, which then reunites with the bulk support material[5,15]. The occurrence of the SMSI state is not restricted to metals supported on titania, with numerous reports describing a wide range of supports and active metals[17–19]. SMSI is, therefore, a fundamental phenomenon for supported metal catalysts.

This rather simplistic picture is currently believed to be true, despite that the atomistic information about the SMSI phenomenon has, for the most part, been characterized ex situ or by means of model systems[14,20]. However, traditional ultra-high vacuum imaging and spectroscopic techniques, such as transmission electron microscopy (TEM) and surface sensitive X-ray photoemission spectroscopy (XPS), can now be performed under relevant in situ conditions[21–25]. The only effort in the 2010s with in situ TEM to gather fundamental new insights into the SMSI state[16] was limited by the use of just a single characterization technique that is not able to capture the multitude of aspects that are important to understand the SMSI state. The aim of the present study was to exploit complementary techniques through a combined investigation of in situ TEM, ambient pressure XPS (APXPS) and in situ powder X-ray diffraction (PXRD) and supported by theoretical density functional theory (DFT) modelling, to derive a holistic view of the SMSI formation mechanism and the role of hydrogen and oxygen within this process. This picture will create a basis to further advance in the use of reductive and oxidative pretreatments to overcome limits of noble metal catalysis.

## Results

**Overlayer formation studied by in situ TEM**. A typical platinum-titania rutile catalyst was synthesized with relatively large platinum particles (~12 ± 6 nm). This particle size enabled diffraction experiments and high-resolution electron microscopy observations. The platinum particles initially exhibited pristine surfaces with no signs of encapsulation (Supplementary Fig. 5). Hydrogen chemisorption measurements after reduction at 200 °C and 600 °C showed a typical decrease[26] in hydrogen chemisorption, from $0.2 \text{ cm}^3 \text{ H}_2 \text{ g}^{-1}$ to $0.1 \text{ cm}^3 \text{ H}_2 \text{ g}^{-1}$. There was no observable sintering of the platinum particles. This is a sign that high-temperature reduction brought the catalyst to the SMSI state. In order to understand this state, we applied a real-space local method in a series of in situ TEM experiments in combination with the integral methods of surface-sensitive APXPS and crystal-structure-sensitive PXRD. The in situ TEM observations (Fig. 1) reported below are general findings that were obtained from the measurements from multiple platinum particles in several experiments. The effect of the electron beam was carefully evaluated by comparing the SMSI state in the region of observation with that in regions that were not exposed to the beam. The catalyst was heated in 1 bar of hydrogen to 600 °C. Upon exposure to hydrogen, a gradual decoration of the platinum surface with an ill-structured material of low-contrast was observed (Fig. 1a, Supplementary Movie 1) and, within minutes, the surface was completely covered (Fig. 1b). Electron energy loss spectroscopy (EELS) mapping at the Ti $L_{2,3}$ edges in scanning transmission electron microscopy (STEM) mode (Fig. 2a, d) revealed these species to be titanium-containing. After changing the gas atmosphere to oxygen—known to destroy the overlayer[15] —the overlayer suddenly increased in volume, accompanied by disturbance of the lattice of the platinum particles (Supplementary Movie 2). The overlayers observed on multiple particles exhibited, in part, a crystalline structure, some disordered areas, as well as small amorphous titanium oxide particles on top of the platinum particle (Fig. 1c, h).

The EELS mapping of the Ti $L_{2,3}$ edge under these conditions (Fig. 2b, d) showed an increase in the titanium signal at the outermost shell of the platinum particle. The titanium spectra of the titania bulk under hydrogen and oxygen were similar, but the spectrum of the overlayer in hydrogen showed an edge position at about ~1 eV lower, suggesting (partial) titania reduction. The gas atmosphere was changed again to hydrogen, after which dynamic changes in the overlayer occurred (Supplementary Movie 3, Fig. 2e). Titania particles as observed in Fig. 1c migrated to a platinum particle edge and shrunk in size, which was accompanied by deformation of the platinum lattice at the platinum particle edge. Over the course of several minutes the overlayer decreased in thickness (Fig. 1d). EELS mapping showed that the surface was enriched with titanium (Fig. 2c). The gas inlet was switched back to oxygen and the reverse process was observed (Supplementary Movie 4). The transition was accompanied by the appearance of structural defects and the reconstruction of the platinum particle (Fig. 2f). Within the temporal resolution of the imaging (6.4 frames per second) these defects and edge-dislocations altered. Low-contrast structures appeared on the particle surface between two captured images. Within seconds after this process, a crystalline layer started to build up on the top surface of the platinum particle. The layer formed from the edge of the particle within 1 s (Fig. 2g). This layer was partially crystalline, consisting of multiple layers of titania. Other particles,

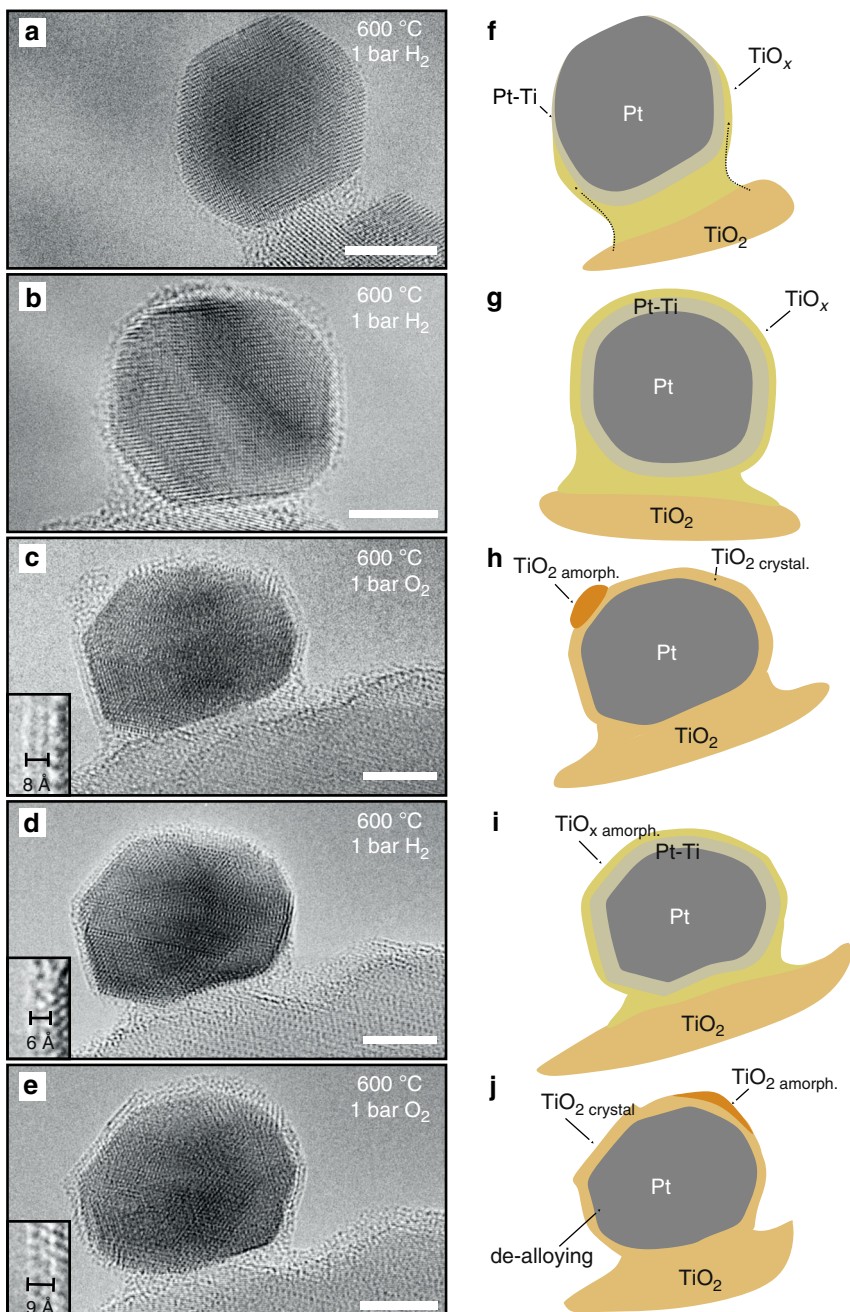

**Fig. 1 Evolution and dynamic structural changes of the overlayer in SMSI.** A platinum particle on a titania support in the first exposure to $H_2$ at 600 °C (**a**, **b**) and the subsequent atmosphere change to $O_2$ at 600 °C (**c**), a switch to $H_2$ (**d**) and then a switch to $O_2$ again (**e**), and interpretation of the phenomena based on the combined results of in situ transmission electron microscopy, in situ X-ray photoemission spectroscopy, and in situ powder X-ray diffraction (**f**–**j**). Insets for **c**–**e** show a magnified image of the overlayer structure observed. Scale bar is 5 nm.

which were not exposed to the electron beam during the experiment, exhibited similar structures indicative of the absence of beam induced artifacts (Supplementary Fig. 5).

**Surface characterized by in situ XPS**. To obtain surface-sensitive chemical data, APXPS was performed on the same catalyst. After reduction at 1 bar $H_2$ at 600 °C and inert transfer to the XPS chamber, the sample was measured at 600 °C in 0.14 mbar $H_2$. The Pt $4f$ spectra showed that platinum was fully reduced (Fig. 3a), displaying an asymmetric $4f_{7/2}$ peak centered at 71.3 eV[27,28]. The subsequent switch to oxygen drastically decreased the Pt $4f$ to Ti$2p$ peak ratio by about 34% of its value under

hydrogen (Fig. 3b), accompanied by a peak shift to a higher binding energy (72.1 eV) and a change in the line shape to Gaussian–Lorentian. The peak position was about 0.8 eV lower in binding energy than for platinum(II) oxide[29]. Such electronic structure of platinum can arise through the formation of Pt-O at the interface of the platinum particle at the titania overlayer, where electron transfer occurs from the platinum to the overlayer. The decrease in intensity of the platinum signal correlated with the formation of the thick encapsulation layer observed by means of in situ TEM (Fig. 1c, e). The switch back to hydrogen led to formation of fully reduced platinum, shown by a shift in the negative binding energy of the Pt from $4f_{7/2}$ to 71.3 eV. Moreover, the intensity of the Pt $4f$ peak increased again (Fig. 3c) to about

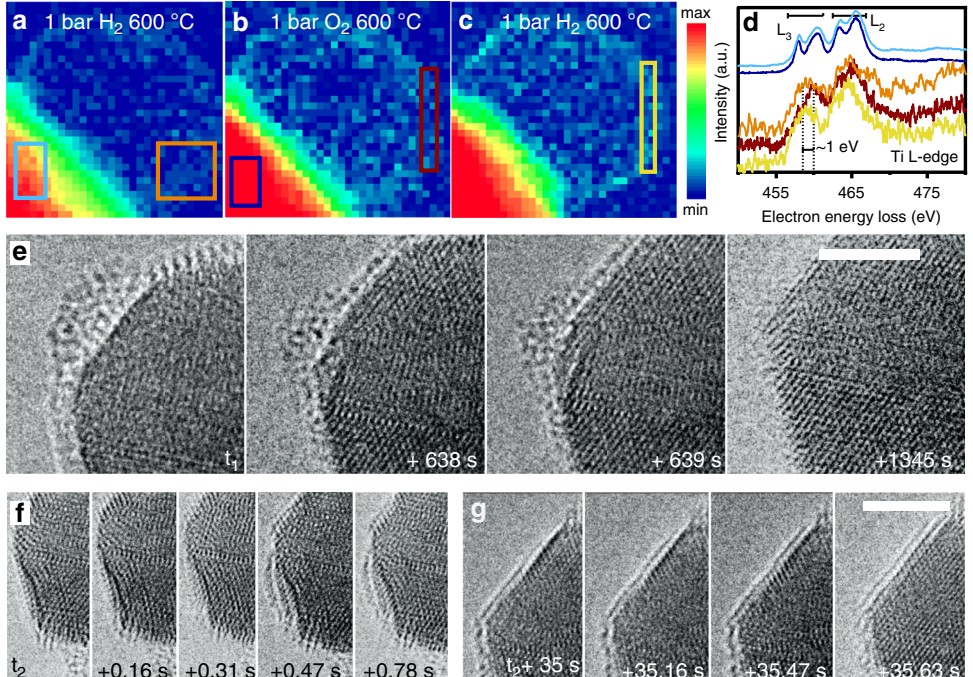

**Fig. 2 EELS mapping and dynamics of metal-support interactions upon gas switches.** EELS mapping between 455–470 eV of EELS spectra of platinum particles on $TiO_2$ in $H_2$ at 600 °C for 1 h (**a**), after a switch to $O_2$ (**b**), and back to $H_2$ (**c**) and the corresponding EELS spectra (**d**) taken from the areas indicated **a**, **b**, and **c**. Dynamic structural changes in the titania overlayer on platinum upon switching the gas atmosphere: from $O_2$ to $H_2$ (**e**) and from $H_2$ to $O_2$ (**f**, **g**). Scale bar represents 5 nm.

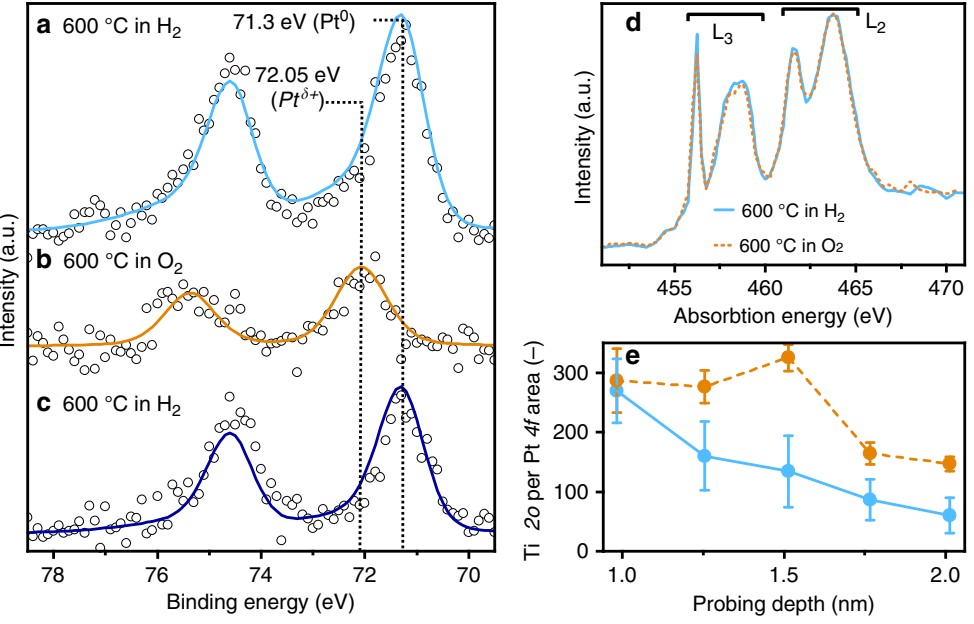

**Fig. 3 In situ spectroscopy.** APXPS (650 eV photon energy) at 600 °C of the Pt $4f$ peak with respective peak fittings (Donjac–Sunjic for **a**, **c** and Gaussian–Lorentian for **b**): after the first $H_2$ (0.15 mbar) exposure at 600 °C (**a**), the Pt signal intensity decreased by 50% after exposure to 1 mbar $O_2$ (**b**); a subsequent switch to $H_2$ (**c**) increased the signal intensity but total intensity remained low. NEXAFS spectra of the Ti $L_{2,3}$ edge (**d**) acquired during the in situ XPS set-up. The second peak ($e_g$) exhibits specific peak shapes for different $TiO_2$ polymorphs and oxidation state. Under $O_2$ at 600 °C characteristics of titania rutile are visible. Under $H_2$ the lack of this feature is characteristic of $Ti_2O_3$ or amorphous titania. Depth profiling (**e**) under $H_2$ at 600 °C (blue) and after the switch to $O_2$ (orange). The probing depth was calculated based on the inelastic mean free path of photoelectrons in titania. The results indicate that a titania layer covered the platinum nanoparticles and that this layer is thicker while dosing oxygen. Error bars in **e** were calculated by Gaussian error propagation of the standard deviation of the Pt $4f$ and Ti$2p$ peak fitting.

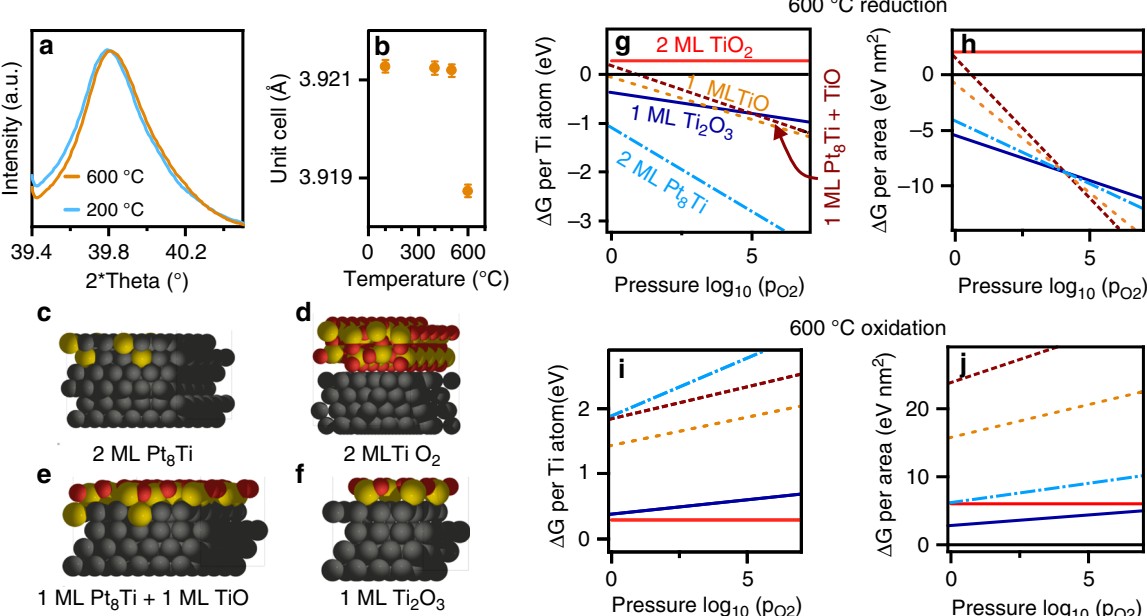

**Fig. 4 Formation of the strong metal-support interaction (SMSI) state.** In situ X-ray diffraction (**a**) acquired at 100 °C at 1 bar in a flow of hydrogen after treatment at two temperatures (200 and 600 °C) and corresponding Pawley fitting for the platinum unit cell after different hydrogen reduction temperatures for 1 h (**b**). The error bars represent one standard deviation of the mean. DFT surface models for different types of platinum surface coverage (**c–f**). The global minimum for the calculated structures at 600 °C as a function of the hydrogen (**g**) and oxygen pressure (**i**) as well as the thermodynamic stability per surface area in the case of a finite number of Ti atoms that may be released by the bulk, as is the case for a kinetic barrier (**h, j**).

60% of its original intensity, in line with the decrease in the encapsulation thickness revealed by TEM (Fig. 1d). The shape of the Ti2p core-level of the X-ray photoemission spectra did not change under the experimental conditions (Supplementary Fig. 6), in agreement with that of stoichiometric titania[30]. The amount of titanium atoms participating in the encapsulation compared to the total amount titanium atoms contributing to the XPS signal was estimated to be below 1% and, thus, is probably below the detection limit of XPS. Electron yield near edge X-ray absorption fine structure (NEXAFS) spectra of the Ti $L_{2,3}$ edge acquired in the same experiment, is extremely sensitive to the local structure around the Ti emitter, while still being surface-sensitive (Fig. 3d). The $e_g$ peak of the $L_3$ edge is indicative of different titania polymorph or reduction state. Under oxygen, the peak structure showed the typical shape of a spectrum of rutile, whereas under hydrogen the fine structure of the $e_g$ peak disappeared, as in the case of $Ti_2O_3$ ($Ti^{3+}$) or due to amorphisation of titania[31,32]. Figure 3e shows the results of an in situ depth profile of the Ti2p and Pt 4f photoemission signals of the sample exposed to different gases at high temperature. In the measurement, the kinetic energy of the photoelectrons was stepwise increased, thus enhancing the depth information of the analysis[33] (Fig. 3e). Measuring at low kinetic energy (330 eV), with the largest surface sensitivity, the Ti to Pt ratio gave approximately the same value, both in hydrogen and in oxygen, because platinum was fully covered by a titanium oxide overlayer with a larger or comparable thickness to the XPS probing depth. With increasing probe depth, the Ti to Pt ratio decreased rapidly, as expected for a thin encapsulation under hydrogen. In the oxygen atmosphere, the depth profile exhibited a less pronounced decrease in the Ti to Pt signal ratio than in hydrogen, proving the greater thickness of the titanium oxide overlayer.

**Pt–Ti alloy formation revealed by in situ XRD.** By in situ TEM, we observed that titanium oxide species moved over the platinum surface in hydrogen. When the gas atmosphere was switched to

oxygen there was a rapid increase in the thickness of the over-layer. This increase in thickness could be solely due to oxidation of the layer or to sequestration and subsequent oxidation of titanium, which formed an alloy with the platinum particle. Therefore, further characterization was carried out to investigate the potential of platinum titanium alloy formation under high-temperature reduction[34], as suggested by the second mechanism for the formation of SMSI. We performed an in situ XRD experiment with the platinum-titania catalyst (Fig. 4a), which showed a shift in the Bragg reflection of the platinum towards higher angles after reduction at 600 °C. Incorporation of titanium into the platinum crystal structure causes lattice contraction, as was observed previously for the $Pt_8Ti$ alloy and for other platinum alloys, which formed upon high-temperature reduction[34–37]. The unit cell contracted from 3.921 to 3.919 Å after reduction at 600 °C, as deduced by means of the Pawley fitting (Fig. 4b).

**Surface structure stability evaluated by DFT.** The stability of various surface models (Fig. 4c–f) was investigated by density functional theory (DFT). In a hydrogen atmosphere at 600 °C, all the reduced overlayers have negative formation energies, thus encapsulation of platinum nanoparticles is thermodynamically favourable. The surface alloy of a $Pt_8Ti$ double-layer on top of platinum was the most stable overlayer when the reduction of titania is a limiting process (Fig. 4g). A mixture of reduced titania and surface alloy was the most stable configuration when the surface area of the platinum nanoparticles is limited (Fig. 4h). However, the alloy double-layer and reduced titania coverage both had only a slightly higher Gibbs free energy. Therefore, potentially all coverage states are thermodynamically possible and are formed in competing manners. Under oxygen at 600 °C (Fig. 4i, j), the surface alloy phases became thermodynamically unfavourable; oxygen-driven de-alloying into a fully oxidized titania phase and a platinum metal surface were the most favourable. A double layer of titania on the metal surface was less favourable by about +0.2 eV with respect to the uncovered

platinum surface. This weak interaction double-layer of TiO₂ with the platinum surface also resulted in a large distance between the two phases in the computational model (Fig. 4d). This data corroborates the formation of titania particles on the metal particle under oxygen.

In this study, the dual role of hydrogen in the formation of the SMSI state and the unique role of oxygen is revealed. Especially the role of oxygen has not been recognized before. We suggest that the removal of the overlayer previously reported after an oxidative treatments stems from other steps applied in these studies. This aspect needs further careful in situ studies. Overall, the presented in situ observations of the prototypical platinum-titania system reveal the complex formation mechanism of overlayers on a platinum-titania catalyst. The migration of reduced titanium oxide onto the platinum particle surface and the formation of an alloy are competing mechanisms during high-temperature reduction. Subsequent exposure to oxygen segregates the titanium from the alloy, and a thicker titania overlayer forms. This thicker fully oxidized overlayer is stable in oxygen and potentially opens new possibilities for catalysis[10]. The stabilization of the oxide overlayer in oxygen environment shows that the SMSI state can as well be utilized in catalytic oxidation reactions. This path so far lacks of intense study. The combination of emerging in situ techniques enables to derive a complimentary and holistic view on materials under conditions that are important for our understanding of phenomena, such as strong metal-support interaction.

## Methods

**Synthesis**. The catalyst was prepared using titanium (IV) oxide Aeroxide® P25 (Acros Organics). The support was impregnated with a solution of tetra-ammineplatinum (II) nitrate (99.995%, Sigma Aldrich) dissolved in ultrapure (MilliQ®) water (incipient wetness impregnation, Pt loading of 2 wt. %). Subsequently the material was calcined for 5 h (heating rate 10 °C min⁻¹) in static air at 200 °C. Then, the catalyst was transferred into a quartz tube, which was heated to 700 °C at 10 °C min⁻¹ in a tubular furnace in a flow of He (50 ml min⁻¹). After 1 h of treatment the tube was cooled to room temperature in He. XRD showed that the anatase phase had disappeared (Supplementary Fig. 1).

**Gas treatment for ex situ TEM**. To compare the results of the in situ experiments, the Pt-TiO₂ sample was also studied with ex situ TEM. A small amount of powder (~5 mg) was loaded between two glass wool plugs and put into a quartz tube. The catalyst was then treated in a flow of O₂ at 50 ml min⁻¹. The sample was heated in a tubular furnace (10 °C min⁻¹) to 600 °C and kept at this temperature for 1 h. Subsequently, the sample was cooled down in a flow of O₂ (50 ml min⁻¹). At room temperature the gas was switched to helium and the sample unloaded.

**Electron Microscopy**. All transmission electron microscopy studies were performed with an aberration-corrected JEOL JEM-ARM300F transmission electron microscope (at 300 kV), which was used at the Scientific Center for Optical and Electron Microscopy (ScopeM) at the ETH Zurich. Electron energy loss spectroscopy (EELS) was performed with a Model 965 GIF Quantum ER Spectrometer/ Imaging Filter. In situ TEM experiments were performed with the DENS solutions "Climate" gas & heating holder and gas feeding system. In situ TEM study was performed according to the profile depicted in Supplementary Fig. 2.

**In situ X-ray photoemission spectroscopy**. In situ XPS measurements were carried out at the X07DB In Situ Spectroscopy beamline (Swiss Light Source, Villigen, Switzerland). The powder samples were dispersed in ultrapure water and drop-casted on silver foil (Aldrich, 99.99% purity). The samples were then treated in a tubular furnace at 600 °C in 1 bar H₂ for 1 h. Then the sample was transferred in inert atmosphere to the beamline and mounted under N₂ on a manipulator and introduced into the solid-gas interface end station, which allows precise dosing of gas under flow conditions[38–40]. Ultrapure gases were dosed by means of mass flow controllers and pumped away with a tunable diaphragm valve connected to a root pump. This allows the dosing of relevant gas flows and a precise control of the pressure during the experiments. The pressure was monitored by means of Baratron measurement heads. The purity of the gases and the switches from oxygen to nitrogen to hydrogen were controlled by a quadrupole mass spectrometer (QMS), located in the second differential pumping stage of the analyzer. The samples were heated using a tunable IR laser (976 nm, max power 25 W), which hit the back of the sample, and the temperature was monitored with a Pt100 sensor. Photoemission spectra were acquired with linearly polarized light, using excitation energy of

650 eV with a pass energy of 20 eV for O, Ti, and C, and 50 eV pass energy for Pt. After alignment of the sample with the photon beam at the focal distance of the analyzer, the sample was investigated by acquiring all the photoemission peaks in a sequence while being exposed to a specific gas at a stable temperature. Schematic of the in situ APXPS experiment is presented in Supplementary Fig. 2.

**XPS peak fitting**. Backgrounds were subtracted using a Shirley background. The peak position was adjusted to the O1s peak at a binding energy of 530.7 eV in order to compensate for a shift in the spectra due to different charging in different gas atmospheres. Peak fitting was performed using the software XPSPEAK41. For the fitting of metallic platinum an asymmetric Donjac–Sunjic[41] line shape was used, with asymmetry parameters of 0.33 and 80. The asymmetry parameters were evaluated on the basis of the Pt 4f peak, obtained from a Pt foil (Supplementary Fig. 3). For the platinum 4f peak under oxygen a Lorentzian-Gaussian (48 % Gaussian, 52 % Lorentzian) fit was used.

**XPS depth profiling**. Depth profiling was performed at a Pass energy of 50 eV for all elements. The inelastic mean free path (IMFP) of the photoelectrons was calculated using the NIST Electron Inelastic-Mean-Free-Path Database[42]. Assumptions for IMFP of TiO₂ rutile: 16 valence electrons, band gap 3.02 eV, density 4.24 g cm⁻³ with using the TPP-2M equation[43]. For platinum the results from Tanuma, Powell and Penn[44] were used. The elemental peaks were normalized according to the following equation:

$$A_{norm} = \frac{A_{peak}}{\#_{sweeps} * \frac{I_{hv} * 3.76}{hv} * \frac{\sigma}{(4\pi)} * (1 + \beta)} \tag{1}$$

with $A_{Peak}$ is the integrated area on the respective Pt or Ti peak, $\#_{sweeps}$ is the number of sweeps per collected spectrum, $I_{hv}$ is the beam photo current at the respective energy (measured after the experiment by means of a photodiode placed in the sample position, and corrected by the quantum yield 3.76/hv), $\sigma$ is the photoionization cross section, and $\beta$ is the influence of asymmetry of the respective elemental orbital at the respective energy. $\sigma$ and $\beta$ were taken from a database[45,46]. Supplementary Table 1 presents parameters of depth profiling.

**Electron yield NEXAFS**. The Ti L-edge NEXAFS measurements were performed at the X07DB beamline, in parallel with the APXPS measurements (same experimental setup). A kinetic energy window ranging from 320 to 340 eV, on the tail of the Ti LMM Auger series, was acquired in the 450–475 eV photon energy range. Such a kinetic energy range was selected to avoid the appearance of photoemission core-level peaks. The area underneath this linear range was proportional to the intensity of the Auger peaks and was used to generate the spectra shown in Fig. 3d.

**In situ powder X-ray diffraction**. Measurements in the in situ diffraction experiments were carried out with a Bruker diffractometer with a Cu X-ray source. There was flow of H₂ (50 mL min⁻¹) through the cell. The heating ramp was 5 °C min⁻¹ for all heating steps. The experiment was performed as depicted in Supplementary Fig. 2. The sample was initially heated to 100 °C and after 1 h pre-treatment at this temperature the X-ray diffraction was recorded. Subsequently, the sample was heated to 200 °C, left at that temperature for 1 h and the cooled to 100 °C. At this temperature the next diffraction was recorded. This procedure was repeated in the same experiment, while changing the reduction temperature to 300 °C, 400 °C, 500 °C, and 600 °C. The refinement of XRD data was performed using TOPAS 6[47]. The Bragg peak at ~40° 2Θ was assigned to Pt–Ti alloy and its shape fitted using Pseudo-Voight function. The position of the peak of the XRD data collected under the different conditions mentioned above was carefully refined with the space group Fm3̄m by Pawley method to obtain the lattice parameters.

**Hydrogen chemical adsorption**. Hydrogen chemisorption measurements were performed with the Micromeritics 3Flex Surface Characterization Analyzer. Before the chemisorption measurements, catalysts were activated according to the following protocol: Around 200 mg of Pt/TiO₂ material were first degassed at 200 °C (5 °C min⁻¹) in vacuum for 1 h. Then, the sample was reduced in a flow of hydrogen at either 200 °C for 1 h or 600 °C for 1 h (heating ramp 5 °C min⁻¹), followed by evacuation at 200 °C for 3 h. H₂ adsorption isotherms were measured at 35 °C. During the measurements, small doses of hydrogen (0.1 cm³ g⁻¹) were equilibrated to acquire the high-resolution adsorption isotherm. After finishing the first adsorption isotherm, the sample was evacuated at 35 °C for 1 h (to remove weakly adsorbed hydrogen) and the measurement of the H₂ isotherm was repeated in order to estimate the amount of physically adsorbed hydrogen.

**Electronic structure method**. DFT calculations were performed using the Quickstep module in the CP2K simulation package[48]. The generalized gradient approximation (GGA), using the Perdew, Burke and Ernzerhof (PBE) functional, was chosen to evaluate the exchange-correlation energy[49]. Valence electrons were treated explicitly, whereas interactions with the frozen atom core were described using Geodecker–Teter–Hutter (GTH) pseudo potentials[50] with 1, 6, 12 and 18 valence electrons for H, O, Ti and Pt, respectively. A double-zeta valence plus polarization (DZVP) basis set, optimized according to the Mol-Opt method[51], was

used. A cut-off of 500 Ry was used for the auxiliary plane wave expansion of the charge density. Brillouin zone integration was performed with a reciprocal space mesh consisting of only the gamma point. The strict convergence criteria of $10^{-7}$ Ha were adhered to in the SCF calculations.

**Ab initio thermodynamic**. The effect of the gaseous environment on the relative stability of the considered surface structure was captured by the method of ab initio atomistic thermodynamics[52]. The Gibbs free energy of a gas phase at temperature T and partial pressure P is given by:

$$G(T, P) = E^{DFT} + E^{ZPE} + \Delta G(T, p^0) + k_B\, T\ln(p/p^0) \qquad (2)$$

where $E^{DFT}$ is the energy calculated by DFT at 0 K, $E^{ZPE}$ is the zero-point energy, $p^0$ is the standard pressure, and $\Delta G(T, p^0)$ includes the contribution from translational, rotational, vibrational and electronic free energy terms of the species under consideration. The detailed derivation for $\Delta G(T, p^0)$ can be found elsewhere[53]. These were implemented in the atomic simulation environment (ASE) Python package[54]. The Gibbs free energy of the solid phase changes with T and p is much smaller than the gas phase and was, thus, neglected in this study. The supported, reduced $TiO_x$ thin oxide overlayer on Pt (111) can have various complex structures, in which the configuration and stoichiometry are very sensitive to the experimental conditions. The TiO rock salt (111) plane and the hexagonal k-phase $Ti_2O_3$ (0001) plane could form a commensurate overlayer on the Pt (111) surfaces and is reported to have a strong interaction with the support. In the present study, the TiO rock salt phase, the $Ti_2O_3$ hexagonal k-phase, rutile $TiO_2$ and $Pt_8Ti$ alloy were used to model the overlayers with different oxidation states ($Ti^{2+}$, $Ti^{3+}$, $Ti^{4+}$ and $Ti^0$).

**Interfacial model—lattice mismatch**. For a given combination of the overlayer/support, $TiO_x$/Pt(111), the interfacial model has three structure variables: The lattice constant of the $TiO_x$ surface, the lattice constant of the Pt(111) surface and the orientation between these two surfaces. It is assumed that the structure of the support does not change significantly due to the presence of the overlayer; thus, a lattice constant of bulk platinum was used and fixed for all interfacial models. The optimal lattice constants of the $TiO_x$ oxide overlayers can differ from that of the bulk oxides, both because of their nanoscopic (ultrathin) character and the oxide/support interaction. Interfacial orientation determines the relative position of the atoms at the interface and, thus, strongly affects the interfacial structure.

Since $TiO_x$ and Pt have different lattice constants, special care is required when constructing atomic models to ensure that the strain is kept to a minimum. Under the periodic boundary conditions, based on the DFT-code, it is possible to build sets of supercells containing varying numbers of overlayers on the support. We assumed a lattice vector of $TiO_x$, defined as $\vec{V}_{TiOx} = m\vec{a}_{TiOx} + n\vec{b}_{TiOx}$, where m and n are integers and $\vec{a}_{TiOx}$ and $\vec{b}_{TiOx}$ are the basis vectors of the primitive cell for $TiO_x$ overlayers. Similarly, the lattice vector of the Pt (111) surface can be defined as $\vec{V}_{Pt} = p\vec{a}_{Pt} + q\vec{b}_{Pt}$, where p and q are integers and $\vec{a}_{Pt}$ and $\vec{b}_{Pt}$ are the basis vectors of the primitive cell for the Pt (111) surface. The lattice constant of the supercell was fixed to be the same as that of the Pt (111) surface. Thus, the lattice mismatch for $TiO_x$ is

$$\Delta = \frac{|\vec{V}_{Pt}| - |\vec{V}_{TiOx}|}{|\vec{V}_{TiOx}|} \qquad (3)$$

The interfacial orientation can be generated by considering rotated $(\sqrt{r} \times \sqrt{r})$-cells (root surface) of either the support metal or the oxide-layer. Supplementary Fig. 4a shows the rotation and lattice constant for some of the root surfaces. Supplementary Fig. 4 gives an example. Considering the computational accuracy, the computational cost and a combination of the lattice constant and the orientation (m, n, p, q and r), a mismatch of $-10\% < \Delta < 10\%$ and a supercell lattice constant smaller than 25 Å were calculated.

**Interfacial model—termination**. Reduced $TiO_X$ overlayers tend to grow according to a quasihexagonal pseudomorphic polar arrangement exhibiting an Pt–Ti–O stacking sequence[55], that is, the Ti atoms at the interface bond with the support and the O atoms above the Ti monolayer. If the lattice constant of the platinum support and the $TiO_x$ oxide overlayer were the same, then the Ti atoms would occupy 50 % of the fcc/hcp hollow sites above the Pt(111) surface, and the O atoms might occupy 50% of the fcc/hcp hollow sites of the Ti monolayer. However, since there is a mismatch between the lattice constants of the two surfaces, rotation plus translation was used to find the optimal position of the overlayers. For rock salt TiO(111)/Pt(111), the most stable interface model is found at a rotational angle at 30°. For hexagonal $Ti_2O_3$ (0001)/Pt(111), the most stable interface model was found at a rotational angle of 0°. For rutile $TiO_2$ (110)/Pt(111) forming supercells with low lattice mismatch is more challenging, since the minimal unit cell of rutile $TiO_2(110)$ is rectangular and that of the Pt(111) hexagonal. However, the rectangular lattice can also be described by a larger rhombic unit cell. We, therefore, generated the rhombic unit cell of rutile $TiO_2$, which is close to the shape of a hexagonal cell. The most stable interface model was found at a rotational angle of 19.1°. For the overlayer of Pt–Ti alloy, $Pt_8Ti$ (space group I4/mmm) was used. The $Pt_8Ti(301)$ surface is the most closely packed and has the same atomic arrangement

as the Pt (111) surface. Thus, the substitution of Pt atoms by Ti atoms on the Pt (111) surface was done to model the overlayer structure.

## Data availability

The authors declare that the data supporting the findings of this study are available within the paper and its Supplementary Information files. High resolution movies and electron microscopy micrographs are openly available in the ETH research collection (https://doi.org/10.3929/ethz-b-000415611). All other relevant data are available from the corresponding authors upon reasonable request.

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

## Acknowledgements

Part of this work was performed at the Swiss Light Source, Paul Scherrer Institute, Switzerland. We acknowledge the Swiss Light Source for providing synchrotron radiation beamtime at the in situ Spectroscopy beamline and the Scientific Center for Optical and Electron Microscopy (ScopeM) at ETH Zurich for the provided electron microscopes. We thank M. Schoenberg for her comments on the manuscript. A.B. and J.A.v.B. acknowledge the SNSF project 200021_178943. X.H. acknowledges the financial support from ETH Career Seed Grant SEED-14 18-2. M.W. and X.H. acknowledge the funding of the SNSF project 200021_181053. X.W. gratefully acknowledges financial support from the China Scholarships Council (No. 201506370019). M.Z. and D.P. thank the ESI platform of Paul Scherrer Institute for financial support. X.W. and D.P. thank the Swiss National Supercomputing Centre (CSCS) for providing the computational resources.

## Author contributions

A.B., X.H., M.W. and J.A.v.B. conceived and planned the research project. A.B. wrote the original and final draft of the paper. X.H. and A.B. performed and analysed the in situ TEM experiments. A.B. and L.A. performed and analysed the APXPS experiments. M.Z. and A.B. carried out chemisorption, synthesis and XRD measurements, P.R. analysed the XRD data. X.W. and D.P. designed and calculated the computational models. All authors contributed to the revision of the paper.

## Competing interests

The authors declare no competing interests.
