## [Peer Review File · Nature Communications]

REVIEWERS' COMMENTS:

Reviewer #1 (Remarks to the Author):

In this manuscript, the authors present new insights into the well-known SMSI phenomenon on Pt/TiO₂ catalysts. The authors used novel in-situ probes, XPS and TEM and coupled with XRD and DFT they present the following picture:

High temperature H₂ reduction causes TiO₂ overlayers to form on the Pt particles, even the large particles studied by the authors here. But in addition to this well-known effect, they also see the formation of a surface Pt-Ti alloy. The evidence for this surface alloy is not that direct, since it is based on the formation of a thicker, crystalline layer of TiO₂ on the surface of the Pt particle after oxidation, when the alloy has been transformed back into Pt. The authors also see a Pt-Ti alloy formation based on the XRD. Now if XRD sees it, then it must be a bulk and not a surface effect. The formation of this alloy is also confirmed by DFT.

In summary, the work sheds new insight into a phenomenon that was studied extensively in the 80s. However, the catalytic significance of this SMSI state continues to be explored and the authors now suggest that we might see an influence of the SMSI even after oxidation. Previously it was thought that oxidation would reverse the SMSI. I am not sure how to reconcile this new observation with the previous work where the chemisorption ability of the catalyst was restored. It would help if the authors could address this minor concern.

Other than that, my only suggestion is to do a careful proofreading since I found some obvious typos and grammatical errors. This was a partial list, the authors should look over carefully.

Line 21 in the abstract "the overlayer was provides.." drop the 'was'

Line 59 "However their understanding Traditional ultra-high vacuum imaging and ..." sentence needs revision.

Line 65 "techniques to a combined.." the word 'to' should be replaced by 'through'

Line 69 "will create to basis.." change 'to' to 'a'

Line 107 "shrunk, which was.." change to 'shrunk in size.'

Line 128 "under hydrogena respectively oxygen." Sentence needs rewording.

Line 135 "ant" replace with 'and'

Reviewer #2 (Remarks to the Author):

The revised version of the ms has significantly improved. I liked the paper already when submitted to [REDACTED], but the presented results are not high-ranking enough to justify publication in [REDACTED]. For Nature Communication the paper is suitable.

Response to the reviewers

Reviewer #1 (Remarks to the Author):

In this manuscript, the authors present new insights into the well-known SMSI phenomenon on Pt/TiO₂ catalysts. The authors used novel in-situ probes, XPS and TEM and coupled with XRD and DFT they present the following picture:

High temperature H₂ reduction causes TiO₂ overlayers to form on the Pt particles, even the large particles studied by the authors here. But in addition to this well-known effect, they also see the formation of a surface Pt-Ti alloy. The evidence for this surface alloy is not that direct, since it is based on the formation of a thicker, crystalline layer of TiO₂ on the surface of the Pt particle after oxidation, when the alloy has been transformed back into Pt. The authors also see a Pt-Ti alloy formation based on the XRD. Now if XRD sees it, then it must be a bulk and not a surface effect. The formation of this alloy is also confirmed by DFT.

In summary, the work sheds new insight into a phenomenon that was studied extensively in the 80s. However, the catalytic significance of this SMSI state continues to be explored and the authors now suggest that we might see an influence of the SMSI even after oxidation. Previously it was thought that oxidation would reverse the SMSI. I am not sure how to reconcile this new observation with the previous work where the chemisorption ability of the catalyst was restored. It would help if the authors could address this minor concern.

We ask this very important question ourselves as well. We are currently addressing this issue in our follow-up research. The original description of the reversibility or restoration of the chemisorption capacity was experimentally done by the following experimental procedure: Low temperature reduction (LTR) = high chemisorption capacity → high temperature reduction (HTR) = low chemisorption capacity → high temperature oxidation → LTR = high chemisorption capacity. Two hypothesis seem to contribute its part to the restoration of the chemisorption capacity a) water that is formed between the switch from O₂ and H₂ for the LTR is efficient in removing the surface overlayer; b) the stability of reduced overlayers decreases with temperature (see our supplementary data). A LTR can as well create mobile enough reduced TiO_x species that reunite with the bulk. Our observations need some extra experimental work, but we expect to be able to elaborate on this issue in depth soon. We added one sentence into the manuscript pointing out, that further research and understanding is necessary.

Other than that, my only suggestion is to do a careful proofreading since I found some obvious typos and grammatical errors. This was a partial list, the authors should look over carefully.

We apologize for these obvious typos and reworked to whole manuscript carefully.

Line 21 in the abstract “the overlayer was provides..” drop the ‘was’

Line 59 “However their understanding Traditional ultra-high vacuum imaging and ...” sentence needs revision.

Line 65 “techniques to a combined..” the word ‘to’ should be replaced by ‘through’

Line 69 “will create to basis..” change ‘to’ to ‘a’

Line 107 “shrunk, which was..” change to ‘shrunk in size.’

Line 128 “under hydrogena respectively oxygen.” Sentence needs rewording.

Line 135 “ant” replace with ‘and’

Reviewer #2 (Remarks to the Author):

The revised version of the ms has significantly improved. I liked the paper already when submitted to [REDACTED], but the presented results are not high-ranking enough to justify publication in [REDACTED]. For Nature Communication the paper is suitable.

We thank both reviewers for the confident and trust in our work.